# Driver Fatigue and Distracted Driving Detection Using Random Forest and Convolutional Neural Network

**Bing-Ting Dong** [1]**, Huei-Yung Lin** [2] **and Chin-Chen Chang** [3,*]

[1] Department of Electrical Engineering, National Chung Cheng University, Chiayi 62102, Taiwan
[2] Department of Computer Science and Information Engineering, National Taipei University of Technology, Taipei 10608, Taiwan
[3] Department of Computer Science and Information Engineering, National United University, Miaoli 36003, Taiwan
[*] Correspondence: ccchang@nuu.edu.tw

**Abstract:** Driver fatigue and distracted driving are the two most common causes of major accidents. Thus, the on-board monitoring of driving behaviors is key in the development of intelligent vehicles. In this paper, we propose an approach which detects driver fatigue and distracted driving behaviors using vision-based techniques. For driver fatigue detection, a single shot scale-invariant face detector (S3FD) is first used to detect the face in the image and then the face alignment network (FAN) is utilized to extract facial features. After that, the facial features are used to determine the driver's yawns, head posture, and the opening or closing of their eyes. Finally, the random forest technique is used to analyze the driving conditions. For distracted driving detection, a convolutional neural network (CNN) is used to classify various distracted driving behaviors. Also, Adam optimizer is used to reinforce optimization performance. Compared with existing methods, our approach is more accurate and efficient. Moreover, distracted driving can be detected in real-time running on the embedded hardware.

**Keywords:** fatigue detection; distraction detection; convolutional neural network; random forest; driving monitoring





## 1. Introduction

Based on the statistics from the Ministry of Transportation and Communications (Taiwan), driver fatigue and distracted driving are the two most common causes of major accidents. In Taiwan, about 20% of traffic accidents each year are due to driver fatigue and distracted driving. Driver fatigue and distracted driving have gradually become the principal causes of road traffic accidents. In some countries, driver fatigue and distracted driving are considered to be as dangerous as drink driving. Some traffic laws also forbid driving for a long period. Therefore, it is crucial to detect both driver fatigue and distracted driving in drivers.

Several studies have formulated methods for detecting driver fatigue and distracted driving [1–4]. These methods can be divided into three categories depending on whether they: (1) use vehicle driving data as recorded by the onboard diagnostic systems; (2) use data on the psychological characteristics of the driver, including electroencephalogram (EEG), electrooculogram (EOG), heartbeat, and finger pulse data; or (3) use vision-based techniques to monitor the driver's status by detecting the driver's yawns, head posture, facial expression, and the opening or closing of their eyes.

In this paper, we propose a monitoring approach which detects driver fatigue and distracted driving, and is based on the random forest approach [3] and a convolutional neural network (CNN) [5]. Our approach does not require the use of invasive techniques to collect data. Therefore, it can be used in practical applications to generate reminders and prevent driving accidents. Figure 1 presents a flowchart of the proposed approach.

The approach detects driver fatigue and distracted driving from images captured by a camera in front of the driver. To detect driver fatigue, the approach first uses a single shot scale-invariant face detector (S3FD) to detect the human face in the image [6], since it performs superiorly for the different scales at different the regions of interest (ROI). The approach then uses the highly accurate face alignment network (FAN) to extract the features of the human face [7]. There are 68 facial feature points extracted from the image, including features such as the eyes and mouth. The defined fatigue parameters are then computed using the extracted facial features. Finally, the random forest is trained to determine whether the driver is fatigued, and whether a warning message should be issued to the driver. To detect distracted driving, we mainly use a data set which we collect ourselves. This data set contains data in seven categories, with six categories reflecting common types of distracted driving and one category reflecting safe driving. A CNN trained on our distracted driving data set is used for testing. A warning signal for distracted driving is generated when some positives are detected in the acquired image sequence.

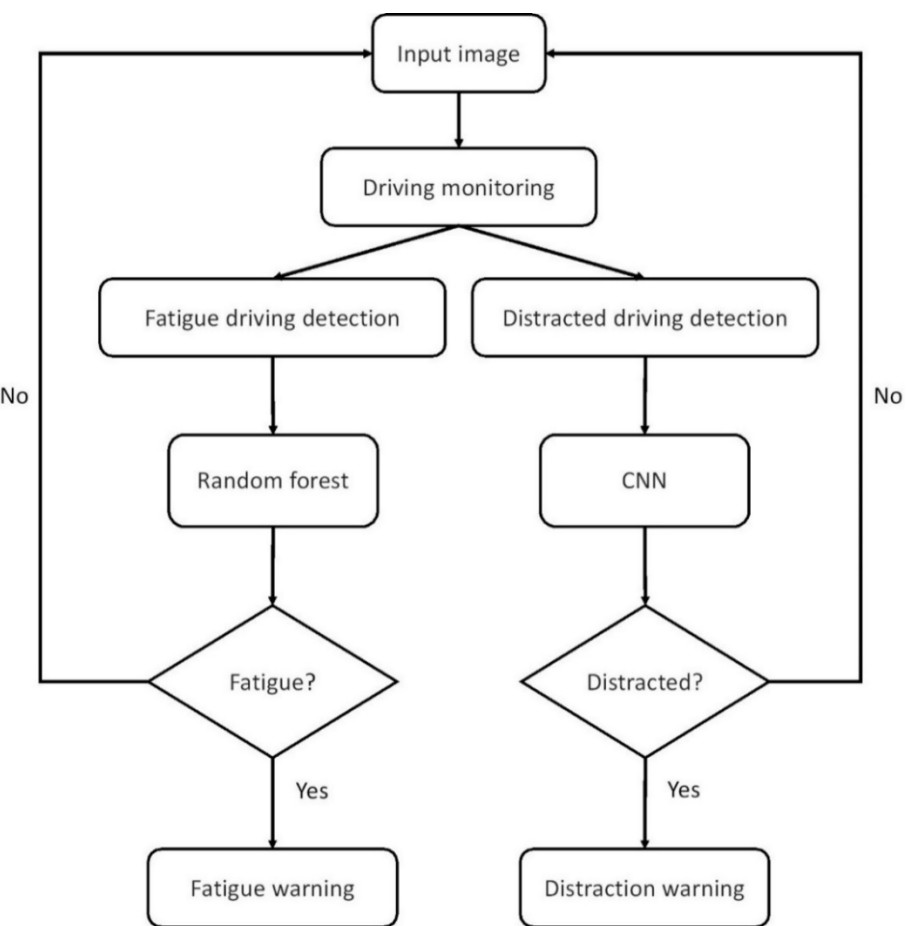

**Figure 1.** Flowchart of the proposed approach for detecting driver fatigue and distracted driving.

Our contributions are two-fold. First, we propose a monitoring approach which performs well in detecting driver fatigue and distracted driving. Driver fatigue detection is based on the random forest approach and distracted driving detection is based on the CNN. A warning message is issued when the driver is fatigued or distracted. Second, our approach can detect distracted driving in real-time by transferring the collected data on the Nvidia Jetson TX2 platform.

## 2. Related Works

In recent decades, various studies have indicated that driver fatigue and distracted driving are the major causes of traffic accidents. Therefore, the on-board monitoring

for driving behaviors has become a critical issue for advanced driver assistance systems (ADAS) in intelligent vehicles.

Kashevnik et al. [4] presented a review of related studies on distracted driving. Distracted driving detection approaches were reviewed and divided into three main types: manual distraction, visual distraction, and cognitive distraction. They also discussed that distracted driving detection approaches can be combined with different algorithms for better performance. Methods that use driving data to determine driver fatigue were primarily based on the movement, acceleration, and steering wheel of the vehicles [8]. However, data in these methods may be inaccurate due to variations in road conditions, vehicle characteristics, and driving styles. By contrast, data from EEG and EOG were objective and reliable [9–12]. Methods using such data can obtain more accurate results than methods using driving data. However, drivers must wear some sensing devices for the physiological data to be obtained. Salvati et al. [11] proposed a detection algorithm for driver fatigue and drowsiness. Physiological signals from the eyes and the heart were extracted and analyzed. Their approach can prevent accidents caused by driver fatigue and drowsiness while driving. However, their approach required non-intrusive devices for collecting physiological signals. Abbas et al. [12] presented a driver fatigue and distracted driving detection technique. They extracted hybrid features using multi-view cameras and biosignal sensors. However, these hybrid features were obtained from non-intrusive devices that drivers must wear, as well as facial expressions.

Vision-based methods were preferred because of their non-contact nature and satisfactory performance [13,14]. The eye closure, yawns, facial expressions, and head posture of the driver were common features used in the analyses [15]. Ou et al. [16] proposed a vision-based technique to detect driver fatigue using image sequences. Their approach was not affected by illumination conditions. However, it did not consider yawning and head posture. Dasgupta et al. [17] used a smartphone to detect driver fatigue. However, they only considered sleepiness-related parameters based on eye motions. Qiao et al. [18] used fatigue-related features, except head posture, in a smartphone-based system for detecting driver fatigue. Galarza et al. [19] introduced an approach to detect driver fatigue using facial expression analysis. Zhang et al. [20] used a CNN to extract spatial image features and a long short-term memory network (LSTM) to analyze temporal features. Their approach had an accuracy rate greater than 87%. Akrout et al. [21] proposed a yawning detection method for driver fatigue based on the spatio-temporal analysis of non-stationary and non-linear signals. They used the YawDD and MiraclHB data sets for evaluation, and the accuracies achieved were 83% and 87%, respectively.

In 2016, State Farm Insurance initiated a distracted driving challenge on Kaggle. The distracted driving data set was the first data set available for training and testing. The competition required the classification of ten driving postures, one reflective of safe driving and nine reflective of distracted driving. Abouelnaga et al. [22] created a new data set (AUC Distracted Driver) similar to the State Farm data set. They applied segmentation to the skin, face, and hand features and proposed a method based on a genetic algorithm. In their approach, five sets of weights for a CNN were used, and the classification accuracy was 95.98%. However, their approach was computationally expensive and thus unsuited to real-time applications. Baheti et al. [23] used the same data set and improved on the VGG-16 network to obtain a classification accuracy of 96.31% when their approach was used in real-time. Kose et al. [24] further improved the classification accuracy for the 10 classes up to 99.10% with real-time processing. In addition, they also combined red-green-blue and optical flow data, and their system performed better than its counterparts on the AUC Distracted Driver and Brain4Cars data sets [25].

Chawan et al. [26] proposed an approach to detect distracted driving. They used three CNN models, namely VGG-16, VGG-19, and InceptionV3. Their approach had a log loss of 0.795. Majdi et al. [27] proposed a supervised learning approach called Drive-Net for detecting distracted driving. Their approach achieved an accuracy of 95%. It was based on a CNN and random forest, and classified representative distracted driving.

Moslemi et al. [28] used a 3D CNN and optical flow with temporal information to improve the detection of distracted driving, and their method achieved an accuracy of 94%. Anber et al. [29] presented a non-invasive algorithm for detecting driver fatigue based on features from the head position and the mouth movements of drivers. Their approach used two pre-trained AlexNet CNN-based models. This approach can obtain a certain detection accuracy, however, testing on a data set with real driving conditions was needed for practical applications.

## 3. Proposed Approach

### 3.1. Driver Fatigue Detection

For the detection of driver fatigue, we use the random forest algorithm because it performs well, has a fast training speed, and can process high-dimensional data. We first detect faces in the input images. Subsequently, we use the S3FD [6] for detecting faces because it can provide high-quality performance for the different scales at different ROIs. Then, we use the FAN [7] to extract 68 facial feature points in the image to find features such as the eyes and mouth. This network uses a 2D representation of the face and the coordinates of the facial feature points as inputs, and it is then trained with four hourglass modules [30]. Hence, upsampling and downsampling can be used to obtain the information of each image size, reduce the loss of image information, and finally acquire a heatmap. This heatmap can be used to predict the position of each facial feature point in the image. The advantage of using this network is that it can detect larger or unusual face poses, and is thus generally more effective than Dlib or the cascaded regression method [31].

For closed eye and eye blink detection, six feature points of each eye are used to define the eye aspect ratio (*EAR*) [32], which is calculated using the length and width of the eye as follows:

$$EAR = \frac{||P_2 - P_6|| + ||P_3 - P_5||}{2(||P_1 - P_4||)} \tag{1}$$

where $P_i$, for $i = 1, 2, \ldots, 6$, are the feature points. A closed eye is defined as $EAR < 0.15$ for a given period. Eye blinking behavior is indicated by the frequent fluctuation of the EAR. For the detection of eye gaze direction, the region of the eye extracted with the six feature points is first converted into a grayscale eye image. The grayscale eye image is then processed with blurring and erosion to eliminate the reflected light, and is finally binarized to derive the enclosing contour and centroid. The eye gaze direction is expressed in terms of the horizontal and vertical directions. Figure 2 illustrates the results from the processing of an eye region extracted with the six feature points.

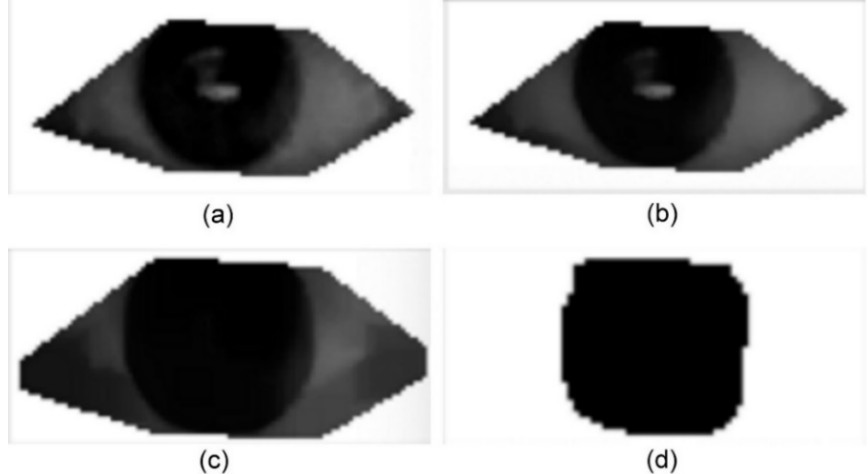

**Figure 2.** Processing results for an eye region. (**a**) grayscale eye image, (**b**) blurred eye image, (**c**) eroded eye image, and (**d**) binarized eye image.

Yawning is such that the mouth is open for a fairly long period. Thus, we use the mouth aspect ratio (*MAR*), similar to the *EAR*, to detect yawning. The *MAR* is defined as follows:

$$MAR = \frac{||P_2 - P_8|| + ||P_3 - P_7|| + ||P_4 - P_6||}{2(||P_1 - P_5||)} \tag{2}$$

where $P_i$, for $i$ = 1, 2, ... , 8, are the eight feature points representing the mouth [33]. A single yawn is indicated by a *MAR* that exceeds a given threshold for a given period (i.e., if the mouth remains open for too long). Figure 3 presents the flowchart for how yawns are detected.

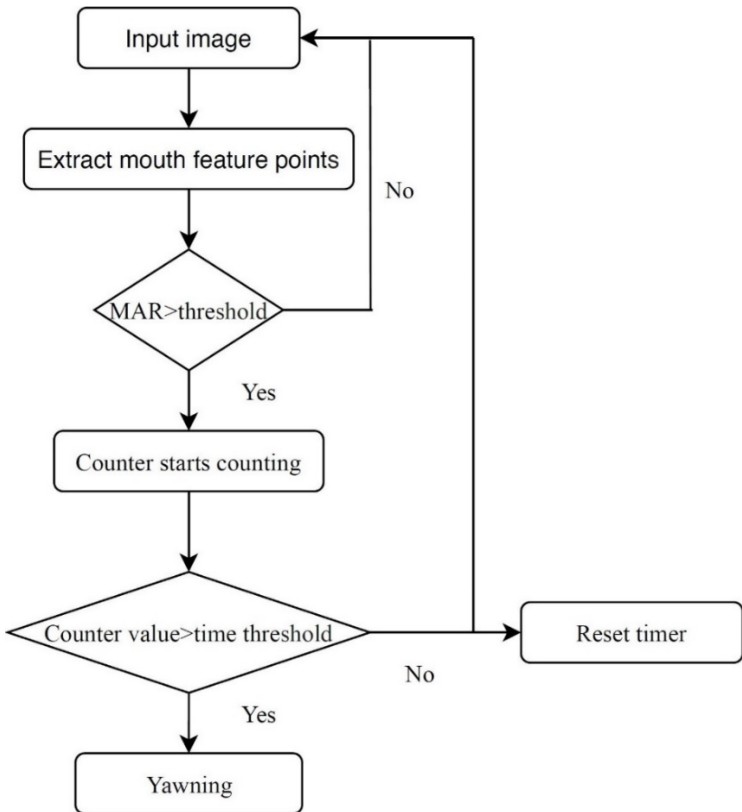

**Figure 3.** Flowchart for detecting yawns.

The purpose of calculating the head posture is to determine whether the driver is nodding. To determine the orientation of the head, we first establish 14 correspondences between the 2D facial feature points and the 3D face model. The identified 2D image features are then mapped onto the 3D model to derive the rotational changes of the head.

For driver fatigue detection, six fatigue parameters are defined: (a) PERCLOS, the percentage of eye closure over time; (b) Blink frequency, the blink frequency in a period of time; (c) Maximum close duration (MCD), the longest eye closure in a period of time; (d) NodFreq, the frequency of nodding over time; (e) YawnFreq, the frequency of yawning in a period of time; and (f) GazeDir, the gaze direction.

We use the YawDD data set for training and testing [34]. This data set contains 322 male and female drivers with or without glasses (or sunglasses). The camera is placed under the front mirror of the vehicle. Three or four video clips are recorded for each participant. Each video shows a different mouth condition, such as normal talking, singing, and yawning. The persons with talking and yawning behaviors are considered as sober and drowsy drivers, respectively. We use the six fatigue parameters to train the random forest, and classify the driver status.

### *3.2. Distracted Driving Detection*

We use a CNN for detecting distracted driving since it is a fast network which performs efficiently. Figure 4 shows the convolutional neural network structure used for distracted driving detection. In the input layer, we first convert the input image to a resolution of $256 \times 256$ pixels. The converted image then passes through five convolutional layers with the activation function ReLU and max pooling. The advantage of using max pooling is that the CNN can run faster. After the convolutional layers, two fully connected layers are combined with ReLU and Dropout. Dropout is included to prevent over-fitting. Finally, the output layer comprises a fully connected layer with a softmax activation function for classification. We use the Adam optimizer for enhanced optimization performance. In the Kaggle distracted driving competition, the log loss was used as an evaluation. Therefore, we use the log function of Log $loss = -(y\log(p) + (1 - y)\log(1 - p))$. In the experiments, a warning signal is generated if a specific distracted driving behavior is detected for 4 out of 10 frames.

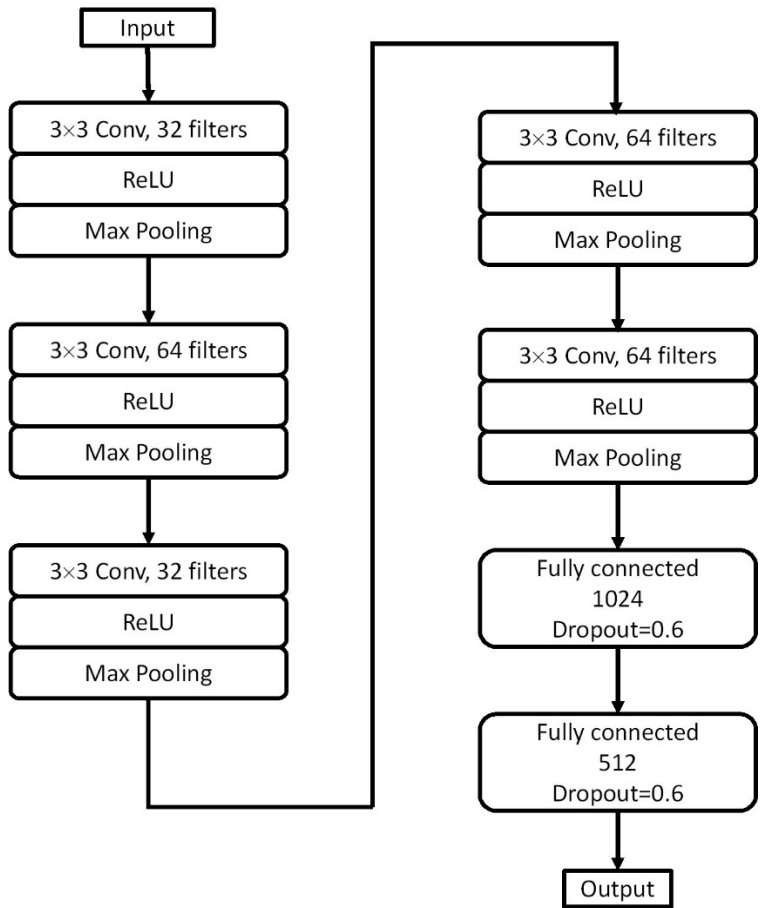

**Figure 4.** Convolutional neural network structure for distracted driving detection.

To collect data on distracted driving, we use two GoPro cameras (with an image resolution of $1920 \times 1080$, horizontal field of view of $95.5°$, and vertical field of view of $56.7°$) to acquire the videos of 12 participants driving. One camera is installed in front of the driver and the other is installed to the right. The data set contains 32,776 images classified into seven categories: "drink", "phone–left", "phone–right", "panel", "texting–left", "texting–right", and 'safe–driving" with 1289, 6217, 5962, 4670, 5982, 6486, and 2160 images, respectively.

## 4. Results

We experimentally tested our approach's ability to detect driver fatigue and distracted driving using the YawDD data set and our own data set. The experiments were performed

on a PC with an Intel i7-8700HQ CPU (Santa Clara, CA, USA) and Nvidia RTX2070 GPU (Santa Clara, CA, USA). Moreover, the experiments were run on the embedded platform equipped with a Nvidia Jetson TX2, dual-core Denver 2 CPU, quad-core ARM A57, and a 256 core GPU (Santa Clara, CA, USA).

### 4.1. Results of Driver Fatigue Detection

We randomly selected five men and five women from the YawDD data set for the experiments. The accuracies of face detection, eye detection, eye opening and closing detection using a CNN, and the eye opening and closing using the EAR metric are shown in Table 1. The average accuracies were 100% for both face and eye detection. Moreover, the average accuracies of eye opening and closing detection using a CNN and the EAR metric were 93.5% and 94.5%, respectively. Hence, the EAR metric performed better than a CNN when used to detect eye opening and closing.

**Table 1.** Detection accuracy. Total: total number of images. Face: face detection accuracy. Eye: eye detection accuracy. O/C CNN: detection accuracy of eye open/close using CNN. O/C EAR: detection accuracy of eye open/close using EAR.

| Video # | Total | Face | Eye | O/C CNN | O/C EAR |
|---------|-------|------|-----|---------|---------|
| 29-male | 644 | 100% | 100% | 89.5% | 94.4% |
| 34-male | 645 | 100% | 100% | 100% | 98.6% |
| 40-male | 638 | 100% | 100% | 100% | 90.1% |
| 42-male | 643 | 100% | 100% | 100% | 93.2% |
| 45-male | 641 | 100% | 100% | 94.6% | 94.3% |
| 12-female | 639 | 100% | 100% | 90.5% | 97.3% |
| 13-female | 779 | 100% | 100% | 90.6% | 96.4% |
| 35-female | 642 | 100% | 100% | 99.3% | 96.3% |
| 37-female | 646 | 100% | 100% | 82.2% | 93.7% |
| 41-female | 432 | 100% | 100% | 86.1% | 87.3% |
| All/Average | 6349 | 100% | 100% | 93.5% | 94.5% |

Figure 5 indicated the detection of eye opening and closing using the CNN and EAR metrics. The orientation of the face was also reflected in the direction of head posture.

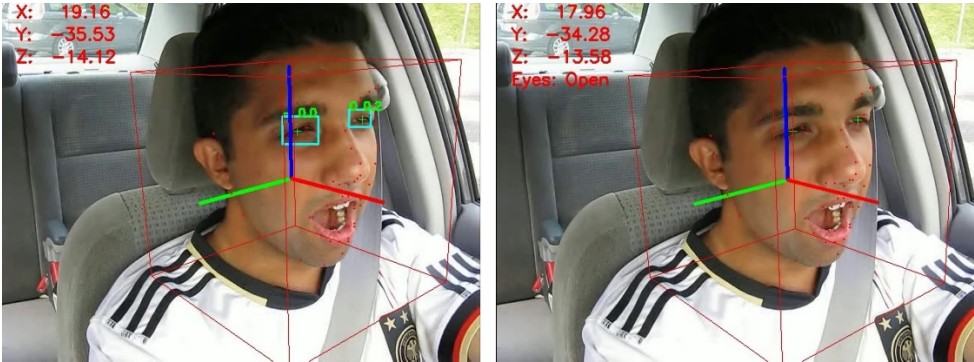

**Figure 5.** Detection of eye opening and closing using a CNN (**left**) and the EAR metric (**right**). The images also indicated the orientation of the face in the detection of head posture.

In the classification of driver fatigue, 20% of all 53 videos in the YawDD data set were used for testing and the rest were used for training. We used the random forest algorithm to classify and analyze driver fatigue. The results are shown in Table 2. The number of trees and the minimum number of samples in a leaf were set as 10 and 1, respectively.

**Table 2.** Results from the random forest algorithm and parameter settings.

|  | Precision | Recall | F1 Score | Support |
|---|---|---|---|---|
| 0 | 0.89 | 1 | 0.94 | 8 |
| 1 | 1 | 0.67 | 0.8 | 3 |
| Accuracy | - | - | 0.91 | 11 |
| Macro average | 0.94 | 0.83 | 0.87 | 11 |
| Weighted average | 0.92 | 0.91 | 0.9 | 11 |

The importance of each fatigue parameter was shown in Table 3. The results revealed that the frequency of yawning was the most crucial parameter for detecting driver fatigue because it was highly related to the use of the YawDD data set for training. The second and third most crucial parameters were blink frequency and PERCLOS, respectively.

**Table 3.** Importance of the fatigue parameters.

| Fatigue Parameters | Importance |
|---|---|
| YawnFreq | 0.387 |
| Blink frequency | 0.149 |
| PERCLOS | 0.130 |
| GazeDir (left) | 0.101 |
| NodFreq | 0.077 |
| MCD | 0.058 |
| GazeDir (center) | 0.055 |
| GazeDir (right) | 0.042 |

We compared our approach in detecting driver fatigue with its counterparts in the literature [20,21,35,36]. Table 4 shows the comparison of our approach with previous methods. At 91%, the proposed approach is more accurate than those of Zhang et al. [20], at 88.6%, Akrout and Mahdi [21], at 83%, Moujahid et al. [35], at 79.8%, and Bakheet and Al-Hamadi [36], at 85.6%.

**Table 4.** Comparison of our approach with the previous methods.

| Method | Accuracy |
|---|---|
| Zhang et al. [20] | 88.6% |
| Akrout and Mahdi [21] | 83.0% |
| Moujahid et al. [35] | 79.8% |
| Bakheet and Al-Hamadi [36] | 85.6% |
| Our approach | 91.0% |

Our approach was also fast. It processed an image in 0.47–0.5 s on the PC (including approximately 0.24 s for the random forest computation) and could output a driver fatigue detection result in 2.8 s on the Jetson TX2 platform.

*4.2. Results of Distracted Driving Detection*

We divided our distracted driving front data set into a training set and a validation set, with 26,223 frames and 6553 frames, respectively. Table 5 shows the confusion matrix for the distracted driving front data set. We evaluated the CNN model, which was trained on data from 11 participants, and on the data from the one remaining participant. The overall training accuracy of the CNN model was 99.7%. The average accuracy, calculated as the number of correct frames divided by the total number of frames, was 91.6%.

**Table 5.** Confusion matrix for the distracted driving front data set.

|     | c0   | c1   | c2   | c3   | c4   | c5   | c6   |
| --- | ---- | ---- | ---- | ---- | ---- | ---- | ---- |
| c0  | 1032 | 0    | 4    | 5    | 0    | 0    | 0    |
| c1  | 0    | 4996 | 1    | 0    | 0    | 0    | 0    |
| c2  | 0    | 0    | 4724 | 0    | 0    | 1    | 0    |
| c3  | 1    | 0    | 2    | 3559 | 0    | 7    | 0    |
| c4  | 9    | 0    | 1    | 0    | 4756 | 1    | 0    |
| c5  | 33   | 1    | 0    | 1    | 0    | 5176 | 0    |
| c6  | 0    | 12   | 0    | 5    | 0    | 0    | 1689 |

Figure 6 presented a resulting image for detecting distracted driving.

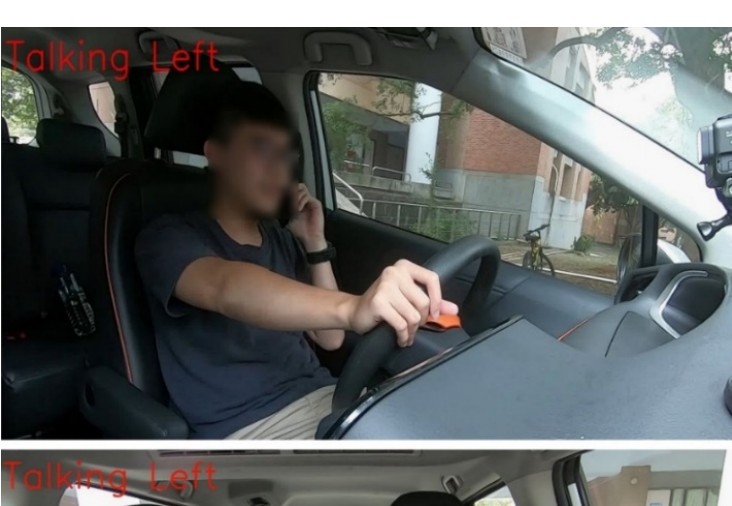

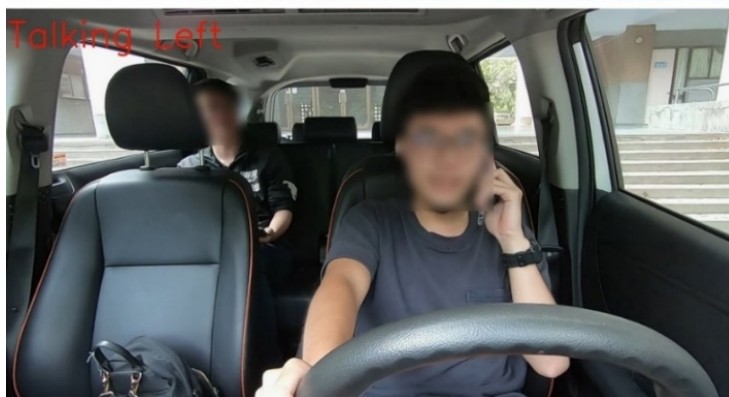

**Figure 6.** Detection of distracted driving.

To compare the distracted driving detection methods, we used the Kaggle distracted driving data set for training and validation. This driving data set was taken by a camera mounted in the car. The data set contained a total of 22,424 training images and 79,726 testing images. The distracted driving behaviors in the Kaggle data set [37] fell into 1 of 10 categories: "safe driving" (c0), "texting–right hand" (c1), "talking on the phone–right hand" (c2), "texting–left hand" (c3), "talking on the phone–left hand" (c4), "operating the radio" (c5), "drink" (c6), "reaching behind" (c7), "hair and makeup" (c8), and "talk to passengers" (c9). For each test image, our approach assigned a probability for each of the 10 categories. We compared the proposed distracted driving detection against several previous methods [27,28,38]. Table 6 shows the comparison of various distracted driving detection techniques. In the proposed approach, the training accuracy and log loss were 98.3% and 0.17, respectively. The accuracy and log loss of the validation set were 97.5% and 0.11, respectively. These results demonstrated the superiority of our approach over existing methods.

**Table 6.** Comparison of the proposed distracted driving detection with previous methods.

| Method | Accuracy |
|---|---|
| Pixel SVC [38] | 18.3% |
| SVC + HOG [38] | 28.2% |
| SVC + PCA [38] | 34.8% |
| SVC + Bbox + PCA [38] | 40.7% |
| VGG-16 [38] | 90.2% |
| VGG-GAP [38] | 91.3% |
| MLP [27] | 82% |
| RNN [27] | 91.7% |
| Drive-Net [27] | 95% |
| Ensemble VGG-16 and VGG-GAP [28] | 92.6% |
| RGB + Optical flow [28] | 94% |
| Our approach | 97.5% |

Table 7 shows the confusion matrix for the Kaggle data set. The frame rates were approximately 140–170 and 40–70 fps for the PC and Jetson TX2, respectively. The results revealed that the proposed approach can achieve real-time detection on the Nvidia Jetson TX2 platform.

**Table 7.** Confusion matrix for the Kaggle data set.

| | c0 | c1 | c2 | c3 | c4 | c5 | c6 | c7 | c8 | c9 |
|---|---|---|---|---|---|---|---|---|---|---|
| c0 | 1949 | 25 | 2 | 17 | 1 | 1 | 0 | 0 | 0 | 0 |
| c1 | 5 | 1799 | 3 | 2 | 0 | 0 | 0 | 1 | 0 | 1 |
| c2 | 0 | 6 | 1819 | 0 | 0 | 4 | 3 | 0 | 11 | 0 |
| c3 | 17 | 2 | 1 | 1805 | 5 | 0 | 0 | 2 | 0 | 0 |
| c4 | 4 | 9 | 2 | 4 | 1819 | 0 | 3 | 10 | 8 | 2 |
| c5 | 6 | 0 | 1 | 3 | 5 | 1826 | 10 | 1 | 1 | 3 |
| c6 | 3 | 2 | 4 | 0 | 0 | 1 | 1849 | 0 | 24 | 0 |
| c7 | 1 | 1 | 0 | 0 | 1 | 0 | 1 | 1577 | 17 | 5 |
| c8 | 0 | 0 | 1 | 0 | 6 | 0 | 1 | 4 | 1523 | 14 |
| c9 | 2 | 0 | 0 | 8 | 0 | 4 | 1 | 2 | 16 | 1673 |

## 5. Conclusions

We have proposed an approach for detecting driver fatigue and distracted driving. Our approach uses the face and posture information of the driver to determine the driving status. For driver fatigue detection, our approach uses six fatigue parameters and the YawDD data set for training. Random forest is then trained to determine whether the driver is fatigued. The average accuracies are 100% for both face and eye detection. Moreover, the average accuracies of eye opening and closing detection using the EAR metric is 94.5%. For classification, the random forest algorithm can achieve the accuracy of 91%. The frequency of yawning is the most crucial parameter for detecting driver fatigue because it was highly related to the use of the YawDD data set for training. Other data sets can be added for training and a smaller feature detection network can be used for improvement. For distracted driving, our approach can be detected in real-time on the Nvidia Jetson TX2 platform. The results demonstrate that our approach performs better than the previous methods.

In future research, we aim to use a CNN to more accurately estimate where the driver is looking and to process eye information for detecting driver fatigue. Moreover, we aim to reduce the computational expense by using a smaller face detection network while maintaining adequate accuracy.

**Author Contributions:** Methodology, B.-T.D. and H.-Y.L.; Supervision, H.-Y.L. and C.-C.C.; Writing—original draft, B.-T.D.; Writing—review & editing, H.-Y.L. and C.-C.C. All authors have read and agreed to the published version of the manuscript.

**Funding:** The authors would like to thank the Ministry of Science and Technology of Taiwan for financially supporting this research under Contract No. MOST 111-2221-E-239-027.

**Institutional Review Board Statement:** Some participants participated in the collection of public databases, and they will be informed by the authors that their facial expressions will be used for research. Some participants participation has been recorded in the author's database, and they will also be informed that their facial expressions will be used for research. The authors have also selected and used a few of their students to be drivers.

**Informed Consent Statement:** Not applicable.

**Data Availability Statement:** Not applicable.

**Acknowledgments:** This paper is a revised and expanded version of a paper entitled, "An on-board monitoring system for driving fatigue and distraction detection," in Proceedings of the 22nd IEEE International Conference on Industrial Technology (ICIT), Valencia, Spain, 10–12 March 2021.

**Conflicts of Interest:** The authors declare no conflict of interest.

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
