# Peer review of "Driver Fatigue and Distracted Driving Detection Using Random Forest and Convolutional Neural Network"

_applsci, doi:10.3390/app12178674_

Round 1
Reviewer 1 Report
1. It is recommended to place the reference sequence in ascending order.
2. Most of the related works are before 2018 (apparently 10 of 18 articles).
3. “HD” should be described the first time it is used (Line 112). Review other acronyms in the document.
4. It is possible to strengthen Figure 1 with the detailed information from section 3.1
5. The description of sections 3.1 and 3.2 should be improved in their wording, some paragraphs seem to be unrelated or the sequence for a smooth reading is not adequate, for example, the relationship between the need to identify the light is not well understood of traffic and the next paragraph.
6. It is recommended to place images and tables immediately once they are cited.
7. The proposal is mainly compared with articles before 2018. It is possible to identify in the literature articles related to the topic of interest and from recent years that could contribute to the research. Still, it is not described if a systematic literature review was carried out to substantiate articles used as related work before 2018.
8. It is recommended to describe ROI the first time it is used.
9. It is recommended to justify, and substantiate in a better way why the selection of the AI techniques used (Random Forest and CNN), a more profound approach is required.
10. It is recommended to review the format and quality of Figures, Tables, and equations.
11. The architecture of the CNN used unconventionally is described, it is recommended to add a table that helps improve the explanation of the network as presented by Lee et al. in his work entitled: Convolutional Neural Network-Based Classification of Driver's Emotion during Aggressive and Smooth Driving Using Multi-Modal Camera Sensors.
12. It is recommended to improve the results section, in the sense of significant progress on your research proposal and the comparison with other related works, as mentioned above, several of the associated works are before 2018.
13. From a general perspective, more rigorous research on related works in the available literature is recommended.
14. It is recommended to better describe the methodology implemented in the research project, which will contribute to a fluid reading and a better understanding of the scientific relevance of the research.
15. It is not possible to infer a significant contribution and innovation when dealing with two problems in the flowchart presented.
16. It is recommended to reorder the structure of the document to identify:
a. Implemented methodology.
b. Comparison between the proposal and related works of recent years.
c. Describe the proposal in a better way to identify through a fluid reading the relevant contribution.
Reviewer 2 Report
Dear authors please update following:
- extend abstract ( please add main aim of the contribution, it is necessary to presented research methods, which were used)
- it is needed to describe and presented scientific methods, which were used in contribution
- please add one paragraph - e.g. current trends / actual state based on surveys, ...
- I do not understand importance of figures 4 and 5. It is necessary more describe them.
- Please extend conclusion and add part of article discussion.
Please improve article, I think, that it is needed to make an effort on this contribution.
Reviewer 3 Report
The paper describes a system for the detection of the fatigue of the driver. The applied techniques are not original and they have been applied to cope with the problem.
The task is interesting since it is related to street security.
No evident motivation is given for the application of two different techniques.
The first tested dataset is not large. The gaggle dataset should be described in a more detailed way.
Minor issues:
Table 3 shows a number of fractional digits that is not necessary
Round 2
Reviewer 2 Report
Dear Authors,
I think, that you provided very good improvements.
Please check English. Some changes are required